# Changes in the Mean of and Variance in Psychological Disease Incidences before and during the COVID-19 Pandemic in the Korean Adult Population

**DOI:** 10.3390/jpm12040576

**Published:** 2022-04-04

**Authors:** So Young Kim, Dae Myoung Yoo, Mi-Jung Kwon, Ji-Hee Kim, Joo-Hee Kim, Woo-Jin Bang, Hyo Geun Choi

**Affiliations:** 1Department of Otorhinolaryngology-Head & Neck Surgery, CHA University, Bundang CHA Medical Center, Seongnam 13496, Korea; sossi81@hanmail.net; 2Hallym Data Science Laboratory, Hallym University College of Medicine, Anyang 14066, Korea; ydm1285@naver.com; 3Department of Pathology, Hallym Sacred Heart Hospital, Hallym University College of Medicine, Anyang 14068, Korea; mulank@hanmail.net; 4Department of Neurosurgery, Hallym University College of Medicine, Anyang 14068, Korea; kimjihee.ns@gmail.com; 5Division of Pulmonary, Allergy, and Critical Care Medicine, Department of Medicine, Hallym Sacred Heart Hospital, Hallym University College of Medicine, Anyang 14068, Korea; luxjhee@gmail.com; 6Department of Urology, Hallym Sacred Heart Hospital, Hallym University College of Medicine, Anyang 14068, Korea; yybbang@gmail.com; 7Department of Otorhinolaryngology-Head & Neck Surgery, Hallym University College of Medicine, Anyang 14068, Korea

**Keywords:** COVID-19, depression, anxiety, panic disorder, schizophrenia

## Abstract

The coronavirus disease 2019 (COVID-19) pandemic has been suggested to increase the risk of depression and anxiety disorders. This study expanded upon previous findings by estimating the changes in medical visits for various psychological disorders during the COVID-19 pandemic compared to before COVID-19. The entire Korean population ≥ 20 years old (~42.3 million) was included. The first COVID-19 case in Korea was reported on 20 January 2020. Thus, the period from January 2018 through to February 2020 was classified as “before COVID-19”, and the period from March 2020 through to May 2021 was classified as “during COVID-19”. Monthly medical visits due to the following 13 psychological disorders were evaluated: depressive disorder, bipolar disorder, primary insomnia, schizophrenia, panic disorder, hypochondriasis, posttraumatic stress disorder (PTSD), anxiety disorder, anorexia nervosa, addephagia, alcoholism, nicotine dependency, and gambling addiction were evaluated. The differences in the number of medical visits and the variance of diseases before and during the COVID-19 pandemic were analyzed using the Mann–Whitney U test and Levene’s test. Subgroup analyses were conducted according to age and sex. The frequencies of medical visits for depressive disorder, bipolar disorder, primary insomnia, panic disorder, hypochondriasis, PTSD, anxiety disorder, anorexia nervosa, addephagia, and gambling addiction were higher during COVID-19 than before COVID-19 (all *p* < 0.001). However, the frequencies of medical visits for schizophrenia, alcoholism, and nicotine dependency were lower during the COVID-19 pandemic than before the COVID-19 pandemic (all *p* < 0.001). The psychological disorders with a higher frequency of medical visits during COVID-19 were consistent in all age and sex subgroups. In the old age group, the number of medical visits due to schizophrenia was also higher during COVID-19 than before COVID-19 (*p* < 0.001). Many psychological disorders, including depressive disorder, bipolar disorder, primary insomnia, panic disorder, hypochondriasis, PTSD, anxiety disorder, anorexia nervosa, addephagia, and gambling addiction, had a higher number of related medical visits, while disorders such as schizophrenia, alcoholism, and nicotine dependency had a lower number of related medical visits during COVID-19 among Korean adults.

## 1. Introduction

Coronavirus disease 2019 (COVID-19) has exposed the worldwide population to both physical and mental health concerns [1,2]. In addition to the physical illness related to the disease, there have been several factors, such as domestic violence, that can induce psychological distress and insomnia during the COVID-19 pandemic [3,4,5,6,7,8]. The latent period, infectivity, and prognosis of COVID-19 cannot be predicted. Thus, quarantine rules have been shifted several times in many countries, which could induce anxiety due to uncertainty. The lack of therapeutics for COVID-19 was another fear during the COVID-19 pandemic. In addition, the emergence of novel variants that evade vaccination-induced immunity discourages the public from receiving vaccines. The long-lasting COVID-19 pandemic impaired social activities and economic status due to social distancing measures and the lockdown of industries, respectively [9]. Thus, it can be suggested that the frequency of psychological disorders increased during the COVID-19 pandemic [10].

A number of recent studies have examined the hazardous impacts of COVID-19 on psychological disorders [8,11,12]. A meta-analysis reported that the pooled prevalence of anxiety and depression was high during the COVID-19 pandemic (33% [95% confidence intervals (CI) = 28–38%] for anxiety and 28% [95% CI = 23–32%] for depression) in the general population [12]. In a global burden of disease study in 204 countries, SARS-CoV-2 infection rates and decreased mobility in humans were related to the increased prevalence of major depressive disorder and anxiety disorder [13]. The impacts of the COVID-19 pandemic on psychological distress were different according to age and sex. For instance, women and younger populations (≤20 years old) showed an increased risk of psychological distress during the COVID-19 pandemic [14]. Our previous questionnaire-based study estimated that there was no significant increase in the rate of depression during the early COVID-19 pandemic period (from 16 August 2020 to 31 October 2020) [15]. However, the prolonged COVID-19 pandemic could increase the rates of depression and other psychological disorders, such as anxiety and posttraumatic stress disorder (PTSD) [16]. Thus, the impacts of the COVID-19 pandemic on a wide range of psychological disorders need to be evaluated in a large population cohort. 

We hypothesized that the COVID-19 pandemic has an impact on various psychological disorders as well as depression and anxiety disorders. To test this hypothesis, the nationwide population data for medical visits due to various psychological disorders, including depressive disorder, bipolar disorder, primary insomnia, schizophrenia, panic disorder, hypochondriasis, PTSD, anxiety disorder, anorexia nervosa, addephagia, nicotine dependency, and gambling addiction, were compared before and during the COVID-19 pandemic.

## 2. Materials and Methods

### 2.1. Ethics

The ethics committee of Hallym University (2021-11-004) approved the use of these data. The study was exempted from the need for written informed consent by the institutional review board.

### 2.2. Participants and Measurement

This study included the entire Korean adult population (~42.3 million, ≥20 years old) without an exception, as a single health insurance system covers the whole country and the entire Korean adult population legally registered to the national health insurance system. Thus, we were able to obtain the data of all Koreans in primary clinics and tertiary hospitals. In this study, we evaluated the frequency of medical visits due to psychological disorders from January 2018 through to May 2021. As the first COVID-19 cases were discovered in Korea on 20 January 2020, and disease prevention and control started on March 2020, we defined the periods of ‘before COVID-19’ and ‘during COVID-19’ as before February 2020 and after March 2020, respectively.

We evaluated the monthly number of medical visits for 13 psychological diseases that are common in primary clinics. The patients were diagnosed with each disease using ICD-10 codes: depressive disorder (F32, F33), bipolar disorder (F31), primary insomnia (F510, G470), schizophrenia (F20, F21, F231, F232, F25), panic disorder (F400, F410), hypochondriasis (F452), PTSD (F431), anxiety disorder (F40, F41), anorexia nervosa (F500, F501, F508), addephagia (F502, F503, F504, F505), alcoholism (F100, F100, F103), nicotine dependency (T652, F170, F171, F172), and gambling addiction (F630, Z726). The medical visits were calculated without duplicate visits, as we had the medical records of the entire hospital or clinics, and patients were identified by unique resident registration numbers.

### 2.3. Statistics

The differences in the mean number of medical visits due to psychological disorders before and during the COVID-19 pandemic were compared using the Mann–Whitney U test for nonparametric values. The differences in the variances of diseases before and during the COVID-19 pandemic were compared using Levene’s test for nonparametric values [17]. For the subgroup analyses, we divided the participants by age (0–19 years old, 20–59 years old, and 60+ years old) and sex.

All analyses were two-tailed, and *p* values < 0.05 were considered to indicate significance. The results were analyzed using SPSS version 22.0 (IBM, Armonk, NY, USA).

## 3. Results

For all examined psychological diseases, there were different frequencies of related medical visits before and during COVID-19 (all *p* < 0.001, Table 1). 

Medical visits due to depressive disorder, bipolar disorder, primary insomnia, panic disorder, hypochondriasis, PTSD, anxiety disorder, anorexia nervosa, addephagia, and gambling addiction were more common during COVID-19 than before COVID-19 (Figure 1A–C,E–J,M). On the other hand, medical visits due to schizophrenia, alcoholism, and nicotine dependency were less common during COVID-19 than before COVID-19 (Figure 1D,K,L). The variance in bipolar disorder and panic disorder was lower during COVID-19 than before COVID-19, while that in gambling addiction was higher during COVID-19 than before COVID-19 (all *p* < 0.05). Other psychological disorders did not show a difference in variance between the two time periods.

Regarding sex, males had a higher frequency of medical visits due to all examined psychological disorders except schizophrenia, addephagia, alcoholism, and nicotine dependency during the COVID-19 pandemic (all *p* < 0.001, Table 2). Medical visits for schizophrenia, alcoholism, and nicotine dependency were less common among males during COVID-19 than before COVID-19 (all *p* < 0.05). Among females, there was a higher frequency of medical visits due to all examined psychological disorders except schizophrenia and nicotine addiction during COVID-19 (all *p* < 0.05). Medical visits due to schizophrenia were less common during COVID-19 than before COVID-19 (*p* = 0.036), and there was no difference in the frequency of medical visits due to nicotine dependency between the two time periods.

Regarding age groups, the 20–39-year-old group demonstrated a higher frequency of medical visits due to all of the examined psychological disorders except for schizophrenia and nicotine dependency (all *p* < 0.05, Table 3). The frequencies of medical visits for schizophrenia and nicotine dependency were lower during COVID-19 than before COVID-19 (both *p* < 0.05). In the 40- to 59-year-old group, there was a higher frequency of medical visits due to all of the examined psychological disorders except schizophrenia, alcoholism, and nicotine dependency during COVID-19; however, there was a lower frequency of medical visits due to schizophrenia, alcoholism, and nicotine dependency during COVID-19 (all *p* < 0.05). In the ≥60-year-old group, there was a higher frequency of medical visits due to all examined psychological disorders except alcoholism and nicotine dependency during COVID-19 than before COVID-19 (all *p* < 0.05).

## 4. Discussion

Various psychological disorders, including depressive disorder, bipolar disorder, primary insomnia, panic disorder, hypochondriasis, PTSD, anxiety disorder, anorexia nervosa, addephagia, and gambling addiction, were more common during the COVID-19 pandemic than before the COVID-19 pandemic in Korean adults. On the other hand, the frequencies of medical visits due to schizophrenia, alcoholism, and nicotine dependency were lower during the COVID-19 pandemic than before the COVID-19 pandemic in Korean adults. Both males and females demonstrated similar trends. In the elderly population, in addition to other psychological disorders, medical visits due to schizophrenia were also more common during the COVID-19 pandemic than before the COVID-19 pandemic. The current results extended previous knowledge on the impact of the COVID-19 pandemic on psychological disorders by addressing the changes in the frequency of medical visits for a wide range of psychological disorders in a nationwide, population-based study.

In the present study, there was an increase in the frequency of medical visits due to depressive disorder, bipolar disorder, primary insomnia, panic disorder, hypochondriasis, PTSD, anxiety disorder, anorexia nervosa, addephagia, and gambling addiction during the COVID-19 pandemic compared with before the COVID-19 pandemic. In line with the current results, many recent studies have demonstrated an increase in the frequency of medical visits due to major psychological disorders, such as depressive disorder and anxiety disorder [18,19,20]. An online survey found that 17.6% of general citizens in 11 countries reported the novel occurrence of PTSD, depression, anxiety, and panic disorder (11.4% reported PTSD, 8.4% reported anxiety, 9.3% reported depression, and 3% reported panic disorder) [20]. The environmental changes during the COVID-19 pandemic and the direct effects of COVID-19 on physical health may have contributed to the increased frequency of medical visits due to these psychological diseases. Quarantine strategies during the COVID-19 pandemic restricted social activities, which could induce social isolation. In addition, nationwide campaigns and information on the COVID-19 pandemic could evoke COVID-19-related fear [21]. Moreover, suffering from COVID-19 could cause physical and mental illness [22]. Among patients with COVID-19, approximately 45% (95% CI = 37–54%) and 47% (95% CI = 37–57%) of patients suffered from depression and anxiety, respectively [15]. It was also suggested that COVID-19 can directly affect the central nervous system, although its predominant target organ is the respiratory tract [23,24,25].

On the other hand, the frequencies of medical visits for schizophrenia, alcoholism, and nicotine dependency were lower during the COVID-19 pandemic than before the COVID-19 pandemic in this study. The impact of the COVID-19 pandemic on medical visits due to schizophrenia has not been well defined. The inaccessibility to medical care could impact the lower number of medical visits for schizophrenia during COVID-19 in this study. Due to the redistribution of medical resources to cope with COVID-19, the shortage of medical access could limit the treatment of schizophrenia during the COVID-19 pandemic. In addition, the deficiency of information and concerns about COVID-19 in patients with schizophrenia could reduce the impact of COVID-19 on psychological symptoms [26]. Patients with schizophrenia have been reported to show less awareness of the COVID-19 pandemic due to poor physical health, poor socioeconomic status, social isolation, and social stigma or discrimination [26,27]. For alcoholism and nicotine dependency, there have been concerns about the increased risk of addiction and dependency on alcohol and nicotine use [28]. However, a cross-sectional survey in an adult population in England demonstrated no significant change in the prevalence of smoking, while attempts to quit smoking increased (adjusted odds ratio = 2.01, 95% CI = 1.22–3.33) [29]. The increased awareness of health care issues and nationwide health care strategies during the COVID-19 pandemic could motivate the cessation of alcohol, smoking, or drug abuse. Additionally, social attention for the cessation of these additions might be decreased while coping with COVID-19; thus, the detection of these addictive behaviors might have been due to the increase in attention.

Among women, in addition to other psychological disorders, the number of medical visits due to alcoholism was higher during the COVID-19 pandemic than before the COVID-19 pandemic in this study. Women have been predicted to be more susceptible to the adverse impact of the COVID-19 pandemic on psychological disorders, such as depression and anxiety [12]. According to age, in the older population, the medical visits due to psychological disorders were higher during the COVID-19 pandemic, even visits due to schizophrenia. The higher prevalence of comorbidities could impose additional vulnerability to various psychological disorders among the elderly population. Social isolation and loneliness might be more common in the elderly population during the COVID-19 pandemic [30], which could result in insufficient socioeconomic support to prevent psychological disorders [31]. It has been previously reported that people who were not supported by sufficient supplies and had low levels of family coherence had a higher incidence of stress, anxiety, and depression [32].

This nationwide population-based study revealed higher incidences of various psychological disorders during the COVID-19 pandemic than before the COVID-19 pandemic. In addition, the current study revealed the unique finding that during the pandemic, the frequencies of medical visits due to some psychological disorders, such as schizophrenia, alcoholism, and nicotine addiction, were lower or not significantly different than before the COVID-19 pandemic. These trends of psychological disorder-related medical visits during the COVID-19 pandemic could be valuable for establishing psychological health policies and managing patients with psychological disorders. However, a few limitations need to be addressed. Because the present study examined the national health claim codes, the population who did not visit the clinic can be missed in this study. In addition, some sensitive or susceptible population could already feel uncomfortable and need mental health care just before COVID-19. Although this study compared the frequency of medical visits due to psychological disorders before and during the COVID-19 pandemic period, the follow-up data for each participant could not be accessed. Thus, causality between the COVID-19 pandemic and psychological disorder-related medical visits was lacking. In addition, comorbid conditions and socioeconomic status could not be considered in this study. Because the COVID-19 pandemic has had an impact on various comorbidities, including cardiovascular diseases, neurologic disorders, and socioeconomic status, changes in these comorbidities and socioeconomic status could mediate the differences in the frequency of psychological disorder-related medical visits. Lastly, this study only analyzed one ethnic population, i.e., Koreans. The severity of the COVID-19 pandemic, the quarantine strategies imposed by a government, and the effects of ethnicity on psychological disorders can influence the frequency of psychological disorder-related medical visits. In Korea, the infection rate of SARS-CoV-2 was lower (less than <1% of the total population) than that in the US or European countries during the study period (until May 2021), and the Korean government maintained stratified social distancing policies, which prevented a lockdown crisis during the COVID-19 pandemic period. Although medical resources were concentrated on COVID-19, medical accessibility was not impaired during the COVID-19 pandemic period. Thus, the trends of medical visits for psychological disorders could be different in other countries.

## 5. Conclusions

The frequencies of medical visits due to many psychological diseases, including depressive disorder, bipolar disorder, primary insomnia, panic disorder, hypochondriasis, PTSD, anxiety disorder, anorexia nervosa, addephagia, and gambling addiction were higher during the COVID-19 pandemic in the Korean adult population. On the other hand, the frequencies of medical visits due to certain psychological diseases, such as schizophrenia, alcoholism, and nicotine dependency were lower during COVID-19 than before COVID-19. The women and old population showed higher susceptibility to the increased psychological diseases during COVID-19 than other populations.

## Figures and Tables

**Figure 1 jpm-12-00576-f001:**
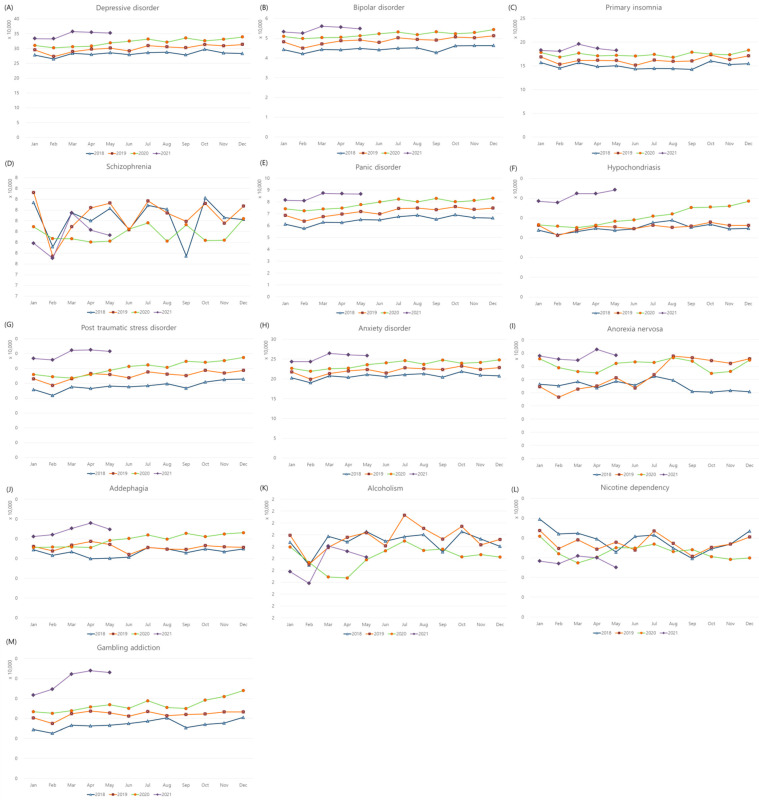
Monthly medical visits for psychological diseases in 2018, 2019, 2020, and 2021. The incidences of (**A**) depressive disorder, (**B**) bipolar disorder, (**C**) primary insomnia, (**D**) schizophrenia, (**E**) panic disorder, (**F**) hypochondriasis, (**G**) posttraumatic stress disorder, (**H**) anxiety disorder, (**I**) anorexia nervosa, (**J**) addephagia, (**K**) alcoholism, (**L**) nicotine dependency, and (**M**) gambling addiction.

**Table 1 jpm-12-00576-t001:** Mean, standard deviation of incidence of diseases before and during COVID-19, and their difference.

Diseases	Before COVID-19		During COVID-19		*p*-Values of Difference	
	Mean	SD	Mean	SD	Mean	Variance
Depressive disorder	292,620.1	13,403.1	331,656.6	15,192.2	<0.001 *	0.065
Bipolar disorder	47,067.7	2728.7	52,970.3	1708.0	<0.001 *	0.046 ^†^
Primary insomnia	157,690.5	9721.9	178,290.8	7332.3	<0.001 *	0.100
Schizophrenia	79,229.4	1583.9	77,719.1	1073.1	<0.001 *	0.113
Panic disorder	68,631.9	4846.0	81,340.6	3953.8	<0.001 *	0.046 ^†^
Hypochondriasis	704.7	35.5	895.8	120.0	<0.001 *	0.224
Posttraumatic stress disorder	2606.5	228.9	3211.4	279.5	<0.001 *	0.140
Anxiety disorder	215,058.6	10,457.6	244,046.6	11,009.8	<0.001 *	0.052
Anorexia nervosa	605.0	95.3	729.1	53.1	<0.001 *	0.080
Addephagia	690.5	43.7	832.1	65.2	<0.001 *	0.108
Alcoholism	19,248.2	482.0	18,489.0	535.9	<0.001 *	0.115
Nicotine dependency	380.4	47.4	308.5	33.6	<0.001 *	0.279
Gambling addiction	297.5	32.3	414.1	69.0	<0.001 *	0.014 ^†^

* Mann–Whitney U test, significance at <0.05. ^†^ Levene’s test in non-parametric data, significance at <0.05.

**Table 2 jpm-12-00576-t002:** Mean, standard deviation of incidence of diseases before and during COVID-19, and their difference in the subgroup by sex.

Diseases	Before COVID-19		During COVID-19		*p*-Values of Difference	
	Mean	SD	Mean	SD	Mean	Variance
**Men**						
Depressive disorder	93,001.0	4416.1	103,448.4	3957.7	<0.001 *	<0.001 ^†^
Bipolar disorder	18,939.4	999.0	20,628.0	537.7	<0.001 *	<0.001 ^†^
Primary insomnia	62,275.8	4002.0	70,463.9	2474.7	<0.001 *	<0.001 ^†^
Schizophrenia	36,100.1	741.9	35,035.8	446.3	<0.001 *	<0.001 ^†^
Panic disorder	32,491.6	2103.5	37,652.8	1529.4	<0.001 *	<0.001 ^†^
Hypochondriasis	367.3	18.9	487.4	88.1	<0.001 *	<0.001 ^†^
Posttraumatic stress disorder	1016.6	84.1	1209.3	97.0	<0.001 *	<0.001 ^†^
Anxiety disorder	88,061.0	4359.1	98,208.9	3824.1	<0.001 *	<0.001 ^†^
Anorexia nervosa	125.0	21.6	147.8	18.1	0.004 *	0.001 ^†^
Addephagia	56.7	7.1	54.6	5.9	0.283	0.352
Alcoholism	15,908.8	388.4	14,929.1	422.6	<0.001 *	<0.001 ^†^
Nicotine dependency	338.4	42.9	267.1	30.0	<0.001 *	<0.001 ^†^
Gambling addiction	284.1	30.6	397.7	68.6	<0.001 *	<0.001 ^†^
**Women**						
Depressive disorder	199,619.1	9033.8	228,208.2	11,259.4	<0.001 *	<0.001 ^†^
Bipolar disorder	28,128.2	1735.0	32,342.3	1176.9	<0.001 *	<0.001 ^†^
Primary insomnia	95,414.7	5747.8	107,826.9	4878.0	<0.001 *	<0.001 ^†^
Schizophrenia	43,129.3	895.2	42,683.3	647.1	0.036 *	0.099
Panic disorder	36,140.3	2746.4	43,687.8	2437.3	<0.001 *	<0.001 ^†^
Hypochondriasis	337.5	24.0	408.4	35.6	<0.001 *	<0.001 ^†^
Posttraumatic stress disorder	1589.9	148.0	2002.1	184.8	<0.001 *	<0.001 ^†^
Anxiety disorder	126,997.6	6125.2	145,837.7	7225.4	<0.001 *	<0.001 ^†^
Anorexia nervosa	479.9	75.5	581.3	41.3	<0.001 *	<0.001 ^†^
Addephagia	633.9	41.3	777.5	64.9	<0.001 *	<0.001 ^†^
Alcoholism	3339.4	115.7	3559.9	147.5	<0.001 *	<0.001 ^†^
Nicotine dependency	42.0	10.4	41.5	7.4	0.947	0.863
Gambling addiction	13.5	3.8	16.4	3.2	0.018 *	0.016 ^†^

* Mann–Whitney U test, significance at <0.05. ^†^ Levene’s test in non-parametric data, significance at <0.05.

**Table 3 jpm-12-00576-t003:** Mean, standard deviation of incidence of diseases before and during COVID-19, and their difference in the subgroup by age.

Diseases	Before COVID-19		During COVID-19		*p*-Values of Difference	
	Mean	SD	Mean	SD	Mean	Variance
**Age 20–39 years old**						
Depressive disorder	71,186.6	8426.6	98,224.07	8281.236	<0.001 *	<0.001 ^†^
Bipolar disorder	16,655.5	1345.4	20,394.40	1039.450	<0.001 *	<0.001 ^†^
Primary insomnia	23,920.1	1123.4	26,040.13	657.012	<0.001 *	<0.001 ^†^
Schizophrenia	25,253.5	535.2	24,169.13	364.634	<0.001 *	<0.001 ^†^
Panic disorder	21,229.2	1928.8	25,956.20	1309.550	<0.001 *	<0.001 ^†^
Hypochondriasis	147.6	13.5	192.33	32.797	<0.001 *	<0.001 ^†^
Posttraumatic stress disorder	1157.4	107.9	1538.40	158.457	<0.001 *	<0.001 ^†^
Anxiety disorder	49,638.9	4329.4	61,145.00	3890.384	<0.001 *	<0.001 ^†^
Anorexia nervosa	151.7	13.4	177.80	17.729	<0.001 *	<0.001 ^†^
Addephagia	507.9	34.0	631.20	51.985	<0.001 *	<0.001 ^†^
Alcoholism	2800.4	142.0	2944.40	124.962	0.003 *	0.002 ^†^
Nicotine dependency	80.0	15.7	67.47	14.918	0.014 *	0.017 ^†^
Gambling addiction	209.5	29.3	304.93	54.928	<0.001 *	<0.001 ^†^
**Age 40–59 years old**						
Depressive disorder	93,126.1	2946.4	98,709.6	3436.8	<0.001 *	<0.001 ^†^
Bipolar disorder	17,747.2	650.9	18,579.3	403.8	<0.001 *	<0.001 ^†^
Primary insomnia	53,514.6	2709.5	58,857.4	1673.9	<0.001 *	<0.001 ^†^
Schizophrenia	39,003.7	913.6	37,193.2	574.2	<0.001 *	<0.001 ^†^
Panic disorder	33,226.0	2009.9	37,970.7	1675.0	<0.001 *	<0.001 ^†^
Hypochondriasis	286.6	16.8	374.6	55.5	<0.001 *	<0.001 ^†^
Posttraumatic stress disorder	949.5	80.0	1072.3	78.5	<0.001 *	<0.001 ^†^
Anxiety disorder	83,498.2	3552.9	90,802.9	3552.5	<0.001 *	<0.001 ^†^
Anorexia nervosa	75.3	11.4	82.5	5.4	0.024 *	0.029 ^†^
Addephagia	136.4	11.7	153.3	16.2	0.002 *	<0.001 ^†^
Alcoholism	9555.7	286.1	8812.7	284.2	<0.001 *	<0.001 ^†^
Nicotine dependency	203.0	32.1	160.9	19.2	<0.001 *	<0.001 ^†^
Gambling addiction	69.5	7.9	87.6	11.9	<0.001 *	<0.001 ^†^
**Age 60+ years old**						
Depressive disorder	128,307.4	3547.4	134,722.9	3962.8	<0.001 *	<0.001 ^†^
Bipolar disorder	12,665.0	770.7	13,996.6	345.2	<0.001 *	<0.001 ^†^
Primary insomnia	80,255.8	6311.4	93,393.3	5484.8	<0.001 *	<0.001 ^†^
Schizophrenia	14,972.2	636.7	16,356.7	485.1	<0.001 *	<0.001 ^†^
Panic disorder	14,176.7	975.5	17,413.7	1064.2	<0.001 *	<0.001 ^†^
Hypochondriasis	270.5	16.0	328.9	36.1	<0.001 *	<0.001 ^†^
Posttraumatic stress disorder	499.5	50.3	600.7	48.3	<0.001 *	<0.001 ^†^
Anxiety disorder	81,921.5	3288.4	92,098.7	3965.8	<0.001 *	<0.001 ^†^
Anorexia nervosa	377.9	83.8	468.9	41.1	0.002 *	<0.001 ^†^
Addephagia	46.2	6.0	47.6	9.0	0.569	<0.001 ^†^
Alcoholism	6892.1	148.7	6731.9	173.4	0.007 *	0.003 ^†^
Nicotine dependency	97.5	14.8	80.1	13.4	0.001 *	0.001 ^†^
Gambling addiction	18.5	3.5	21.5	4.3	0.023 *	0.020 ^†^

* Mann–Whitney U test, significance at <0.05. ^†^ Levene’s test in non-parametric data, significance at <0.05.

## Data Availability

Restrictions apply to the availability of these data. Data were obtained from the Korean National Health Insurance Sharing Service (NHISS) and are available at https://nhiss.nhis.or.kr (accessed on 25 January 2022) with the permission of the NHIS.

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
