# Peer review of "Changes in the Mean of and Variance in Psychological Disease Incidences before and during the COVID-19 Pandemic in the Korean Adult Population"

_jpm, 2022, doi:10.3390/jpm12040576_

Round 1
Reviewer 1 Report
The study has some merits, including (i) a population-based sample; (ii) a comparison between pre-COVID-19 pandemic and during COVID-19 pandemic; (iii) a variety of psychological distress; and (iv) visualization of the trend in psychological distress among South Korean. There are some improvements that should be made. Please see my comments below.
1. In Introduction, more information and citations are needed. (i) it is suggested that domestic violence could be increased during the COVID-19 pandemic (Ashraf et al., 2021) and this may be also contribute to psychological distress during COVID-19; (ii) four systematic reviews have quantified the issues of psychological distress and insomnia during COVID-19 pandemic, they are relevant to the Introduction (Alimoradi et al., 2021a, 2021b; Olashore et al., 2021; Rajabimajd et al., 2021); (iii) the authors mentioned that "However, the prolonged COVID-19 pandemic could increase the rates of depression and other psychological disorders, such as anxiety and posttraumatic stress disorder (PTSD)." But there is no citation to support. Maybe the article of Lu et al. (2021) is relevant. However, the authors are advised to search literature and find other relevant references to support this statement; (iv) the authors said "Furthermore, we thought the real number of medical clinic visits might be different from the results based on the survey." What makes the authors have this thoughts?
References:
Ashraf A, Ali I, Ullah F. Domestic and gender-Based violence: Pakistan scenario amidst COVID-19. Asian J Soc Health Behav 2021;4:47-50
Alimoradi, Z., Gozal, D., Tsang, H. W. H., Lin, C.-Y., Broström, A., Ohayon, M. M., & Pakpour, A. H. (2022). Gender-specific estimates of sleep problems during the COVID-19 pandemic: Systematic review and meta-analysis. Journal of Sleep Research, 31(1), e13432.
Alimoradi, Z., Broström, A., Tsang, H. W. H., Griffiths, M. D., Haghayegh, S., Ohayon, M. M., Lin, C.-Y., Pakpour, A. H. (2021). Sleep problems during COVID-19 pandemic and its’ association to psychological distress: A systematic review and meta-analysis. EClinicalMedicine, 36, 100916.
Lu, M.-Y., Ahorsu, D. K., Kukreti, S., Strong, C., Lin, Y.-H., Kuo, Y.-J., Chen, Y.-P., Lin, C.-Y., Chen, P.-L., Ko, N.-Y., Ko, W.-C. (2021). The prevalence of posttraumatic stress disorder symptoms, sleep problems, and psychological distress among COVID-19 frontline healthcare workers in Taiwan. Frontiers in Psychiatry, 12, 705657.
Olashore AA, Akanni OO, Fela-Thomas AL, Khutsafalo K. The psychological impact of COVID-19 on health-care workers in African Countries: A systematic review. Asian J Soc Health Behav 2021;4:85-97
Rajabimajd N, Alimoradi Z, Griffiths MD. Impact of COVID-19-related fear and anxiety on job attributes: A systematic review. Asian J Soc Health Behav 2021;4:51-5
2. In Methods, I would suggest having a wash-out period (e.g., between December 2019 to February 2020). Say, the COVID-19 happened from a minor stage, and some sensitive people may already feel uncomfortable and need mental health service.
3. It is unclear to me if the authors treated each participant as independent in the Methods. That is, if a participant sought help from hospital three times and another sought help five times, will they be counted as three and five, or will they be counted as one and one? This is important for readers to interpret the data reported in Tables. Moreover, have the authors adjusted for time? That is, whether the periods were the same for before COVID-19 and during COVID-19 when the authors counted the frequencies.
4. In the Discussion, the authors should discussed one possibility that some vulnerable people (e.g., people with schizophrenia) are advised not to visit the hospital. This may contribute to the low frequency. Some vulnerable people themselves may also not want to visit hospital. Also, some restriction policies may also contribute to the visits.
5. According to the previous comment, do the authors have information to further separate the during COVID-19 period into "strong prevention control (e.g., lockdown)" and "weak prevention control (e.g., can free mobility with good personal hygiene)" by the governments?
Author Response
The study has some merits, including (i) a population-based sample; (ii) a comparison between pre-COVID-19 pandemic and during COVID-19 pandemic; (iii) a variety of psychological distress; and (iv) visualization of the trend in psychological distress among South Korean. There are some improvements that should be made. Please see my comments below.
- In Introduction, more information and citations are needed. (i) it is suggested that domestic violence could be increased during the COVID-19 pandemic (Ashraf et al., 2021) and this may be also contribute to psychological distress during COVID-19; (ii) four systematic reviews have quantified the issues of psychological distress and insomnia during COVID-19 pandemic, they are relevant to the Introduction (Alimoradi et al., 2021a, 2021b; Olashore et al., 2021; Rajabimajd et al., 2021); (iii) the authors mentioned that "However, the prolonged COVID-19 pandemic could increase the rates of depression and other psychological disorders, such as anxiety and posttraumatic stress disorder (PTSD)." But there is no citation to support. Maybe the article of Lu et al. (2021) is relevant. However, the authors are advised to search literature and find other relevant references to support this statement; (iv) the authors said "Furthermore, we thought the real number of medical clinic visits might be different from the results based on the survey." What makes the authors have this thoughts?
References:
Ashraf A, Ali I, Ullah F. Domestic and gender-Based violence: Pakistan scenario amidst COVID-19. Asian J Soc Health Behav 2021;4:47-50
Alimoradi, Z., Gozal, D., Tsang, H. W. H., Lin, C.-Y., Broström, A., Ohayon, M. M., & Pakpour, A. H. (2022). Gender-specific estimates of sleep problems during the COVID-19 pandemic: Systematic review and meta-analysis. Journal of Sleep Research, 31(1), e13432.
Alimoradi, Z., Broström, A., Tsang, H. W. H., Griffiths, M. D., Haghayegh, S., Ohayon, M. M., Lin, C.-Y., Pakpour, A. H. (2021). Sleep problems during COVID-19 pandemic and its’ association to psychological distress: A systematic review and meta-analysis. EClinicalMedicine, 36, 100916.
Lu, M.-Y., Ahorsu, D. K., Kukreti, S., Strong, C., Lin, Y.-H., Kuo, Y.-J., Chen, Y.-P., Lin, C.-Y., Chen, P.-L., Ko, N.-Y., Ko, W.-C. (2021). The prevalence of posttraumatic stress disorder symptoms, sleep problems, and psychological distress among COVID-19 frontline healthcare workers in Taiwan. Frontiers in Psychiatry, 12, 705657.
Olashore AA, Akanni OO, Fela-Thomas AL, Khutsafalo K. The psychological impact of COVID-19 on health-care workers in African Countries: A systematic review. Asian J Soc Health Behav 2021;4:85-97
Rajabimajd N, Alimoradi Z, Griffiths MD. Impact of COVID-19-related fear and anxiety on job attributes: A systematic review. Asian J Soc Health Behav 2021;4:51-5
Response: The recommended references were added.
We agree to your comments. The indicated sentence was removed.
- In Methods, I would suggest having a wash-out period (e.g., between December 2019 to February 2020). Say, the COVID-19 happened from a minor stage, and some sensitive people may already feel uncomfortable and need mental health service.
Response: We agree to your advice. The limitation was added in the discussion section.
“In addition, some sensitive or susceptible population could already feel uncomfortable and need mental health care during just before COVID-19.”
- It is unclear to me if the authors treated each participant as independent in the Methods. That is, if a participant sought help from hospital three times and another sought help five times, will they be counted as three and five, or will they be counted as one and one? This is important for readers to interpret the data reported in Tables. Moreover, have the authors adjusted for time? That is, whether the periods were the same for before COVID-19 and during COVID-19 when the authors counted the frequencies.
Response: Because all Korean population legally registered to the national health insurance system with the identity code, there was little risk on the redundancy of data. Each participant was counted as one even though he was visited clinics for many times.
The time duration for the follow-up was Jan 2018 through February 2020 for “before COVID-19” and March 2020 through May 2021 for “during COVID-19”. This was described in the method section.
In this study, we evaluated the frequencies of medical visits due to psychological disorders from Jan 2018 through May 2021. As the first COVID-19 cases were discovered in Korea on January 20, 2020, and disease prevention and control started on March 2020, we defined the periods of ‘before COVID-19’ and ‘during COVID-19’ as before February 2020 and after March 2020, respectively.
To adjust time variable, the monthly incidence psychological diseases were collected in this study. This was described in the method section.
“We evaluated the monthly number of medical visits for 13 psychological diseases that are common in primary clinics.”
- In the Discussion, the authors should discussed one possibility that some vulnerable people (e.g., people with schizophrenia) are advised not to visit the hospital. This may contribute to the low frequency. Some vulnerable people themselves may also not want to visit hospital. Also, some restriction policies may also contribute to the visits.
Response: The recommended points were described in the discussion section.
“The inaccessibility to medical care could impact the lower number of medical visits for schizophrenia during COVID-19 in this study. Due to the redistribution of medical resources to cope with COVID-19, the shortage of medical access could limit the treatment of schizophrenia during the COVID-19 pandemic. In addition, the deficiency of information and concerns about COVID-19 in patients with schizophrenia could reduce the impact of COVID-19 on psychological symptoms.[19] Patients with schizophrenia have been reported to show less awareness of the COVID-19 pandemic due to poor physical health, poor socioeconomic status, social isolation, and social stigma or discrimination.[19, 20]”
- According to the previous comment, do the authors have information to further separate the during COVID-19 period into "strong prevention control (e.g., lockdown)" and "weak prevention control (e.g., can free mobility with good personal hygiene)" by the governments?
Response: In Korea, the incidence of COVID-19 was lower than other countries during the study period, and there was no complete lockdown period. This was described in the discussion section.
“In Korea, the infection rate of SARS-CoV-2 was lower (less than < 1% of the total population) than that in the US or European countries during the study period (until May 2021), and the Korean government maintained stratified social distancing policies, which prevented a lockdown crisis during the COVID-19 pandemic period. Although medical resources were concentrated on COVID-19, medical accessibility was not impaired during the COVID-19 pandemic period. Thus, the trends of medical visits for psychological disorders could be different in other countries.”
Reviewer 2 Report
Line 94: change could gather to were able to obtain
If the data was collected by the authors, keep the sentence as is. If the data was obtained from the clinics and hospitals, then change as suggested above.
Author Response
Line 94: change could gather to were able to obtain
If the data was collected by the authors, keep the sentence as is. If the data was obtained from the clinics and hospitals, then change as suggested above.
Response: The phrase was revised as advised.
Reviewer 3 Report
Dear authors,
this is an important, new study.
My main recommendation to you is to enrich the study with relevant international references from other countries which have been published at MDPI. It is important to highlight the discussion part. In addition, the conclusions' part need to be developed a little further summarizing the main points- this part is too brief.
note: eg at the special issue of Sustainability at MDPI there are similar articles about covid 19 pandemic effects
https://www.mdpi.com/2071-1050/14/3/1576
from special issue 14/3: https://www.mdpi.com/2071-1050/14/3
Author Response
this is an important, new study.
My main recommendation to you is to enrich the study with relevant international references from other countries which have been published at MDPI. It is important to highlight the discussion part. In addition, the conclusions' part need to be developed a little further summarizing the main points- this part is too brief.
note: eg at the special issue of Sustainability at MDPI there are similar articles about covid 19 pandemic effects
https://www.mdpi.com/2071-1050/14/3/1576
from special issue 14/3: https://www.mdpi.com/2071-1050/14/3
Response: The recommended reference was added.
The conclusion was revised.
Round 2
Reviewer 1 Report
The authors have satisfactorily addressed all my prior concerns. I believe that the present submission is good for publication.